# Analytical and Experimental Study of Fatigue-Crack-Growth AE Signals in Thin Sheet Metals

**DOI:** 10.3390/s20205835

**Published:** 2020-10-15

**Authors:** Roshan Joseph, Victor Giurgiutiu

**Affiliations:** Department of Mechanical Engineering, University of South Carolina, Columbia, SC 29208, USA; victorg@sc.edu

**Keywords:** structural health monitoring (SHM), acoustic emission (AE), fatigue crack growth, high cycle fatigue (HCF) experiment, AE signal mode separation, predictive modeling

## Abstract

The acoustic emission (AE) method is a very popular and well-developed method for passive structural health monitoring of metallic and composite structures. AE method has been efficiently used for damage source detection and damage characterization in a large variety of structures over the years, such as thin sheet metals. Piezoelectric wafer active sensors (PWASs) are lightweight and inexpensive transducers, which recently drew the attention of the AE research community for AE sensing. The focus of this paper is on understanding the fatigue crack growth AE signals in thin sheet metals recorded using PWAS sensors on the basis of the Lamb wave theory and using this understanding for predictive modeling of AE signals. After a brief introduction, the paper discusses the principles of sensing acoustic signals by using PWAS. The derivation of a closed-form expression for PWAS response due to a stress wave is presented. The transformations happening to the AE signal according to the instrumentations we used for the fatigue crack AE experiment is also discussed. It is followed by a summary of the in situ AE experiments performed for recording fatigue crack growth AE and the results. Then, we present an analytical model of fatigue crack growth AE and a comparison with experimental results. The fatigue crack growth AE source was modeled analytically using the dipole moment concept. By using the source modeling concept, the analytical predictive modeling and simulation of the AE were performed using normal mode expansion (NME). The simulation results showed good agreement with experimental results. A strong presence of nondispersive S0 Lamb wave mode due to the fatigue crack growth event was observed in the simulation and experiment. Finally, the analytical method was verified using the finite element method. The paper ends with a summary and conclusions; suggestions for further work are also presented.

## 1. Introduction

Engineering structures are prone to various types of failures during their operation. The failure mechanism during their operation depends on the mechanism of the external loading, working conditions, the microstructural changes inside the structure, etc. Various kinds of damages in metallic structures include fatigue, friction, static failure, etc. It is important to detect the damage initiation in its early stage to prevent catastrophic failure. Structural health monitoring (SHM) is an emerging technology with multiple applications in the evaluation of critical structures. Acoustic emission (AE) is a passive SHM method having a lot of potential for early-stage detection of damages and prevention of the formation of the complete failure of the structure. In the AE method, there are several approaches for damage detection, of which physics-based approaches are receiving a lot of attention these days due to their comprehensive scope. In physics-based approaches, a fundamental understanding of the AE source is necessary to identify the correct AE signals and separate them from the noise and the signals due to confounding factors (environment, temperature changes, friction in joints, etc.) [1].

Analytical modeling of the AE source has also been performed by many researchers [2]. The theoretical studies and experiments were performed to understand the underlying principles, the source mechanism, and used those understandings for simulations. These studies helped to understand the AE signal signature and source mechanism in both theoretical and experimental perspectives. A generalized theory of AE and its source mechanism representations for a half-space were developed by Ohtsu and Ono [3]. Ohtsu and Ono further continued the work by performing the simulation of tensile crack and shear crack in half-space [4]. This paper describes how significant the source mechanism is in forming the signal characteristics. The analytical models consider the source as a self-equilibrating seismic moment tensor due to buried microcrack or dislocations [5]. However, these studies focused on half-space applications such as earthquake seismology. Many researchers also reported analytical studies of AE in plate structures. Weaver and Pao [6] studied the response of an infinite elastic plate to dynamic loading using the method of superposition of normal modes. In this research, Pao studied the numerical results of the surface response of a plate at different locations from the source for different modes of the Rayleigh-Lamb spectrum. Gorman and Prosser [7] studied the normal mode solution to the classical plate bending equation for its applicability to AE. An experimental and theoretical comparison for a pencil lead break in a plate was presented in this paper. Lysak [8] presented a generalized investigation of growing crack acoustic emission from the standpoint of fracture mechanics. An analytical relationship between crack parameters and AE signal parameters was obtained in this research. AE source modeling in plates for various modes of fracture was also studied by researchers [9]. In a recent analytical study of AE guided wave propagation in a plate [10], the AE source was modeled as Helmholtz excitation potentials.

Numerous studies were done to solve the AE forward problem by utilizing numerical finite element method (FEM) models [11,12,13,14]. Several studies considered the AE source as either a monopole or a dipole point source [15]. Finite element modeling works performed by Hamstad, Sause, and Prosser showed how the Lamb wave propagation in plates happens due to an AE activity [11,15,16]. Hora and Cervena investigated the difference between nodal sources, line sources, and cylindrical sources to build geometrically more representative acoustic emission sources [17].

In general, the FEM AE models in metallic structures considered dipole or monopole AE source definitions according to the microcrack orientation. Multiple dipoles in orthogonal directions represent the fiber breakage and matrix crack source representations in the case of composite material AE source definitions. These models require definition of the source dipoles/monopoles, as well as the force-time curves. Some researchers recently modeled the crack growth by local degradation of the material stiffness utilizing a cohesive zone element approach for AE prediction [18]. Cuadra et al. [19] presented an experimental data-driven computational approach to simulate crack initiation as an acoustic emission source directly. In this research, two modeling approaches, including cohesive zone element and Extended Finite Element Method (XFEM) finite element formulations, were implemented leveraging experimental mechanical and full-field nondestructive evaluation information provided by the digital image correlation method. Other researchers used peridynamic formulation withhomogenization and mapping between elastic and fracture parameters of the microscale peridynamic bonds and the macroscale parameters of the composite [20].

AE signals travel as guided waves in thin metallic plates. Therefore, the modeling of AE in thin metallic plates can be approached through the guided wave modeling approach. The analytical modeling of guided waves is very useful for quick numerical predictions for simple geometries [21,22,23]. The FEM method has been proven as a very reliable method for modeling guided wave propagation. However, FEM simulations use extensive computational time. Many numerical techniques were developed to solve this problem. Finite difference equations [24], semi-analytical finite element method (SAFE) [25,26], boundary element method [27], global matrix method (GMM), spectral finite element method [28,29], combined global analytical local FEM analysis [30], analytical global-local method [31], hybrid SAFE-GMM method [32], etc. were developed to improve the computation efficiency of guided wave propagation.

In metallic structures, the AE technique has been used for fatigue crack growth damage detection and source localization. The AE due to fatigue crack growth, as well as wave scattering from fatigue cracks [33,34,35,36,37,38], has been studied by many researchers. The acoustic emission signatures of fatigue damages in an idealized bevel gear spline and two different AE signal signatures for plastic deformation and crack jump were identified by Zhang et al. [39]. The AE signal signatures recorded by piezoelectric wafer active sensor (PWAS) transducers during a fatigue crack growth event in thin metallic plates were studied by Bhuiyan et al. [40,41,42]. In this research, under a slow frequency of fatigue loading (<0.25 Hz), for a short advancement of crack length, the AE signals were recorded, and eight signal signatures were discovered related to crack growth and crack rubbing and clapping. AE signals during fatigue crack rubbing/clapping [43] and vibration-induced fatigue crack rubbing/clapping were also studied experimentally and analytically [44]. The correlation between acoustic emission count rates and crack propagation rates was studied by Roberts and Talebzadeh [33]. Signal processing methodologies and clustering techniques have been used by researchers to differentiate AE signals due to various activities [45]. Different techniques have also been developed for the correlation between crack characteristics and AE signal features [46,47,48,49].

The AE signal features such as duration, rise time, counts, and frequency contents are strongly dependent on the characteristics of the physical phenomenon that generated the AE signal [50]. Several studies have utilized AE measurements to identify the corresponding fracture mechanisms [51,52]. Time-frequency domain-based approaches and modal separation approaches were used for this purpose. The time-domain features of AE measurements are strongly affected by experimental conditions, whereas frequency content is not changed. Thus, the frequency spectrum of AE signals can give more reliable information about the physical phenomenon that produced the AE signals [53].

Many researchers used time-frequency representations of the AE signals for distinguishing the signal types. Suzuki et al. [54], used the wavelet transform for analyzing the AE signals from a longitudinal glass-fiber-reinforced composite sample under tensile loading. The authors analyzed the AE signals recorded using piezo patches and classified the signals into four types using the wavelet transform method. Correlation of signal types to Mode-I fiber fracture, Mode-I matrix crack, and Mode-II disbonding was also performed in this research. Many researchers proved the time-frequency transform technique as a very useful tool for AE signal analysis [55,56,57]. In the present paper, we use both frequency spectrum analysis and time-frequency decomposition analysis of AE signals for AE signal characterization.

The present paper discusses the predictive modeling of the fatigue crack growth AE signals in thin sheet metals. The novelty of the present research is the derivation of a closed-form solution for predictive modeling of fatigue crack growth AE signals sensed using PWAS sensors in thin plates and its experimental validation. The paper starts with the theoretical principles of modeling the AE signals according to the instrumentation used and the working principles of PWAS sensors. Then, it discusses a brief review of the in situ AE experiment and the results. Subsequently, the paper derives the theoretical expression for predictive modeling of PWAS sensed AE signals due to a fatigue crack growth event in thin metallic plates, followed by a comparison with the experimental results. Finally, the paper verifies the analytical solution derived from a comparison with FEM results. The final part of the article presents the summary and conclusions, and makes suggestions for further work.

## 2. AE Modeling Methods

This section discusses the modeling methods used for AE prediction. Due to a fatigue crack source, an acoustic wave is generated. The wave transmits through the metallic plate, recorded by the AE instrumentation. During the transmission from the source to the digitization at the data acquisition (DAQ), the wave passes through a few pieces of equipment. The acoustic signals are modified when they pass through the instrumentation according to their specifications. These transformations happening to the acoustic signals are discussed in this section.

### 2.1. AE Signal Flow Diagram

The AE signal flow diagram from the initiation at the source to the digitization at the DAQ is presented in Figure 1. The sudden dynamic disturbance at the AE source location causes the generation of AE waves. In thin metallic plates, the AE waves travel as Lamb waves. The AE waves propagate in the plate and then reach the PWAS transducer, resulting in a strain field. The strain field at the transducer causes it to generate a response in the form of voltage. The voltage sensed by the transducer is then magnified when it reaches the pre-amplifier. Lastly, the amplified signal is recorded by the DAQ as a digitized signal.

The block diagram of the signal transformations happening during the signal acquisition process is explained in Figure 2. When the AE signal passes through each element of the AE signal acquisition system, a transformation according to the frequency response of the element happens. Due to an AE source such as a fatigue crack growth in thin plates, the generation of AE waves happens according to the structural transfer function of the source on the plate. In thin-walled plates, the AE waves travel as Lamb waves. When the AE waves reach the PWAS sensor, the Lamb waves are modified by the PWAS transfer function or PWAS tuning, which depends on the Lamb wave mode. A signal amplifier was used between the DAQ and the PWAS. The signal amplifier used in the present experiment was a MISTRAS 2/4/6 pre-amplifier, which is a bandpass filter from 30 to 700 kHz. The DAQ system used has a bandwidth of 0–2 MHz. When the signal passes through each element, a modification happens according to the transfer function/bandwidth of the element.

### 2.2. PWAS Transfer Function

In Section 2.1 we have discussed the transformations happening in the AE signal during each stage in the signal flow path. The signal amplifier and DAQ have a specific bandwidth to transfer the signal. The signal reception by PWAS depends on the tuning of the PWAS transducer on the metallic plate. The tuning of the Lamb waves by the PWAS depends on the Lamb wave mode. A closed-form expression for PWAS sensing of Lamb waves is derived in this section. The PWAS senses the in-plane strain caused by the AE waveform and converts it into an equivalent voltage. A schematic of the sensing of Lamb waves by a PWAS sensor is presented in Figure 3.

The interaction between electrical and mechanical variables can be described by linear relations for linear piezoelectric materials. The tensorial form of constitutive relation between electrical and mechanical variables is established as follows:(1)$$E_{i}=-g_{ikl}\sigma_{kl}+\beta_{ik}^{T}D_{k},$$
(2)$$S_{ij}=s_{ijkl}^{E}\sigma_{kl}+g_{kij}D_{k},$$
where *E_i_* is the electric field, *g_ikl_* is the piezoelectric voltage coefficient, *σ_ikl_* is the stress on the PWAS, *β^T^_ik_* is the impermittivity coefficient, *D_k_* is the electric displacement, *S_ij_* is the strain on the PWAS, and *s^E^_ijkl_* is the compliance.

Considering the one-dimensional (1D) PWAS sensing assumption, we get
(3)$$S_{1}=s_{11}^{E}\sigma_{1}+g_{31}D_{3},$$
(4)$$E_{3}=-g_{31}\sigma_{1}+\beta_{33}^{T}D_{3}.$$

Eliminating stress between equations, we get
(5)$$g_{31}S_{1}-s_{11}^{E}\beta_{33}^{T}D_{3}=g_{31}^{2}D_{3}-E_{3}s_{11}^{E}.$$

Upon rearranging, we get
(6)$$g_{31}S_{1}+E_{3}s_{11}^{E}=(g_{31}^{2}+s_{11}^{E}\beta_{33}^{T})D_{3},$$
where
(7)$$D_{3}=\frac{C_{e}V_{3}}{A},$$
(8)$$E_{3}=-\frac{V_{3}}{t_{a}},$$
where *C_e_* is the external capacitance, *V*_3_ is the voltage, *A* is the area of the PWAS, and *t_a_* is the thickness of the PWAS.

For infinitesimal area *dA* and voltage *dV*, substituting Equations (7) and (8) into Equation (6), we get
(9)$$g_{31}S_{1}dA-\frac{dV_{3}}{t_{a}}s_{11}^{E}dA=(g_{31}^{2}+s_{11}^{E}\beta_{33}^{T})\frac{C_{e}dV_{3}}{dA}.$$
Upon rearranging, we get
(10)$$\frac{g_{31}S_{1}dA}{\left(g_{31}^{2}+s_{11}^{E}\beta_{33}^{T}\right)}-\frac{s_{11}^{E}dV_{3}dA}{\left(\frac{g_{31}^{2}}{\beta_{33}^{T}}+s_{11}^{E}\right)\beta_{33}^{T}t_{a}}=C_{e}dV_{3},$$
where
(11)$$\frac{dA}{\beta_{33}^{T}t_{a}}=C_{c},\;\mathrm{the}\;\mathrm{capacitance}\;\mathrm{of}\;\mathrm{PWAS}.$$

Substituting *C_c_* into Equation (10), we get
(12)$$\frac{g_{31}S_{1}}{\left(g_{31}^{2}+s_{11}^{E}\beta_{33}^{T}\right)}-\frac{C_{c}s_{11}^{E}dV_{3}}{\left(\frac{g_{31}^{2}}{\beta_{33}^{T}}+s_{11}^{E}\right)}=C_{e}dV_{3}.$$
Upon rearranging, we get
(13)$$dV_{3}=\frac{g_{31}^{}S_{1}dA}{\left(C_{e}+\frac{s_{11}^{E}\beta_{33}^{T}C_{c}}{g_{31}^{2}+s_{11}^{E}\beta_{33}^{T}}\right)\left(g_{31}^{2}+s_{11}^{E}\beta_{33}^{T}\right)}.$$

The external capacitance is usually very small compared to the capacitance of the PWAS, i.e., *C_e_ << C_c_*. Considering a very small value of *C_e_*, we get
(14)$$dV_{3}=\frac{g_{31}^{}}{\left(s_{11}^{E}\beta_{33}^{T}C_{c}\right)}S_{1}dA.$$

This voltage is due to the strain at an infinitesimal area *dA*. The total voltage sensed by the PWAS can be obtained by integrating the voltage over the area *S_c_.* Thus, we get the total voltage sensed by PWAS as follows:(15)$$V_{3}=\frac{g_{31}^{}}{\left(s_{11}^{E}\beta_{33}^{T}C_{c}\right)}\int_{S_{c}}S_{1}dA.$$

## 3. Review of AE Experiments

### 3.1. Specimen Preparation

For capturing AE during fatigue crack growth in thin metallic plates, an experimental specimen was designed. A commonly used aircraft material, aluminum 2024-T3, was chosen for preparing the test specimens. A shear metal cutting machine was used for preparing coupons of 103 mm width, 305 mm length, and 1 mm thickness from a large plate of aluminum 2024-T3. Such a wide specimen was designed for allowing a long crack formation in the specimen. First, a 1 mm hole was drilled at the geometric center of the specimen. Then, fatigue loading in an MTS machine was applied to the specimen for generating a pre-crack of 4 mm tip-to-tip length. The maximum and minimum loads applied for the cyclic fatigue loading were 13.85 kN and 1.38 kN. Then, the specimen was taken out of the MTS machine, and the PWAS sensor was installed at a distance of 25 mm from the pre-crack. Then, a nonreflective clay boundary (NRB) was implemented on the specimen to prevent the reflection of the AE signals from the boundaries. PWAS sensors were bonded using M-Bond AE-15 adhesive, which is resilient to the disbonding of PWAS during extended durations of cyclic loading. M-Bond AE-15 is an epoxy system with two components: a resin and a curing agent. The resin and curing agent were mixed in a specific proportion and stirred for 5 min. This mixture was used to bond the PWAS to the specimen. The PWAS was bonded to the specimen using the mixture and cured for 3 h at 60 °C as recommended by the epoxy manufacturer. The capacitance of the PWAS was measured before bonding and after bonding. The capacitance values made sure that the PWAS, as well as the bonding, was defect-free. The NRB was applied to the specimen to prevent AE signal reflections from the plate boundaries and, thus, to receive reflection-free, clean AE signals. After the AE sensor and NRB implementation on the specimen (Figure 4), the cyclic fatigue loading was continued. The crack was grown an additional 5.4 mm (until the crack length reached 9.4 mm tip to tip), simultaneously capturing the AE signals. The wide geometry of the specimen was desired for this work so that the acoustic waves generated would travel a longer distance to the edges. This hypothesis, in turn, means the signals die out due to geometric spreading and material damping before reaching the sensor after the reflection from the boundaries. The properties of the Al 2024-T3 specimen were as follows: modulus of elasticity 73 GPa, density 2767 kg/m^3^, and Poisson’s ratio 0.33.

### 3.2. AE Experimental Set-Up

After installing the sensor and NRB, the fatigue loading was continued to grow the crack and capture AE signals simultaneously. The experimental set-up for capturing the AE signal from a fatigue crack growth event is presented in Figure 5. The test specimen installed with the PWAS transducer was mounted on the MTS machine. The bond quality assurance of PWAS sensors was performed periodically by electromechanical impedance spectroscopy (EMIS). The acoustic pre-amplifier connected to the PWAS sensor is a bandpass filter, with a filtering range between 30 kHz to 700 kHz. The pre-amplifier operates with either a single-ended or differential sensor provided with 20/40/60 dB gain (can be selected using a switch). A 40 dB gain was selected in the present experiment. The pre-amplifier was connected to the MISTRAS AE system. For capturing any high-frequency AE signals, a sampling frequency of 10 MHz was chosen. The timing parameters set for the MISTRAS system were as follows: peak definition time (PDT)= 200 µs, hit definition time (HDT) = 800 µs, and hit lockout time (HLT) = 1000 µs.

### 3.3. Experimental Results

In this experiment, the AE signals during fatigue crack growth events recorded using PWAS were identified. Two examples of the AE signals are presented in Figure 6a,b. The frequency spectrum of the signals is also presented in Figure 6c,d. As we observe, the time-domain signal has a sharp rise in the beginning. The frequency spectrum of the signal also has specific peaks and valleys in both signals, which are comparable to each other. A detailed discussion of the results of the experiment is presented in Joseph and Giurgiutiu [58].

The group velocity dispersion curve of a 1 mm aluminum plate is presented in Figure 7. The S0 and A0 Lamb wave modes travel at different velocities with the S0 Lamb wave mode at a higher velocity than the A0 mode. The group velocity dispersion curve of 1 mm aluminum is superimposed in the time-frequency plot of the AE signal in Figure 6e,f. From the time domain, as well as the time-frequency domain representation superimposed with the group velocity dispersion curve of the AE signal, we observe the strong presence of S0 mode in the experimentally observed AE signal. This observation proves that the AE signals from fatigue crack events travel as Lamb waves.

## 4. Predictive Modeling of Fatigue Crack AE

### 4.1. Predictive Modeling of Fatigue Crack Growth AE

In this section, the analytical predictive modeling of AE signals due to the fatigue crack growth process is presented. The schematic of the AE source modeling approach for a fatigue crack growth event is presented in Figure 8. Figure 8a shows the fatigue specimen with a pre-crack and the fatigue loading applied to the specimen. A schematic of the zoom-in of the fatigue crack tip is presented in Figure 8b. During the fatigue crack growth, the crack propagation was observed to happen in the *z*-direction (creating a fracture surface in *y–z* plane). Therefore, the normal to the fracture surface is the *x*-direction. The fatigue loading applied during the experiment was also in the *x*-direction, as we observe from Figure 8a. A fracture in which the loading and normal to the fracture plane are in the same direction is called a Mode-I fracture. Thus, a fatigue crack growth event in the present experimental set-up is considered as a Mode-I fracture process from the definitions of fracture mechanics.

The edge view of the fatigue fracture process is presented in Figure 8c. Assuming a 1D waveguide, and considering that the Lamb wave is decoupled from the shear horizontal waves, one can represent the effective excitation due to such a Mode-I fracture as an *M*_11_ dipole moment excitation, as we observe in Figure 8c.

Therefore, the *M*_11_ dipole moment component excitation represents a Mode-I fracture process in a 1D waveguide and the equivalent of the excitation happening during fatigue crack growth event. Next, for the predictive modeling of the AE, using a closed-form solution of the wavefield due to the dipole moment, excitation needs to be derived. The wavefield due to a concentrated moment excitation is derived from the wavefield due to a concentrated force following the approach mentioned by Aki and Richards [5]. The method is discussed in short below.

Suppose a couple moment is generated due to a force vector *Q* applied at the position ξ(α,β) and another force vector *Q* applied at $\xi^{\prime }(\alpha ,\beta +\Delta X_{j})$ in the opposite direction to that of the force applied at $\xi^{\prime }(\alpha ,\beta +\Delta X_{j})$, where $\Delta X_{j}$ is a small distance in the $X_{j}$ direction as shown in Figure 9.

The displacement field at an arbitrary point *x* due to a point force *Q* at an arbitrary point ξ(α,β) and *Q* at $\xi^{\prime }(\alpha ,\beta +\Delta X_{j})$, as shown in Figure 9, is denoted as $u_{i}^{\xi }$ and $u_{i}^{\xi^{\prime }}$, respectively. If the material is linearly elastic and the displacement field has a linear relation with force applied, we can write
(16)$$u_{i}^{\xi }=Q_{i}G(x;\alpha ,\beta ),$$
(17)$$u_{i}^{\xi^{\prime }}=Q_{i}G(x;\alpha ,\beta +\Delta X_{j}),$$
where *G* is the Green function for a point force. The displacement field at *x* due to a couple with the moment $M_{ij}=Q_{i}\Delta X_{j}$ can be written from Equations (16) and (17) as follows:(18)$$u_{i}^{M_{ij}}=u_{i}^{\xi }-u_{i}^{\xi^{\prime }}=Q_{i}G(x;\alpha ,\beta )-G(x;\alpha ,\beta +\Delta X_{j}),$$
(19)$$=Q_{i}\Delta X_{j}\frac{G(x;\alpha ,\beta )-G(x;\alpha ,\beta +\Delta X_{j})}{\Delta X_{j}}.$$

For an infinitesimally small distance $\Delta X_{j}$, we apply the limits to get
(20)$$u_{i}^{M_{ij}}=\underset{\left(\begin{array}{l} \Delta X_{j}\to 0\\ Q_{i}\to \infty \\ Q_{i}\Delta X_{j}=M_{ij} \end{array}\right)}{Limit}Q_{i}\Delta X_{j}\frac{G(x;\alpha ,\beta )-G(x;\alpha ,\beta +\Delta X_{j})}{\Delta X_{j}}.$$

Using the first-order differential formula, Equation (20) can be written as
(21)$$u_{i}^{M_{ij}}=\underset{\left(\begin{array}{l} \Delta X_{j}\to 0\\ Q_{i}\to \infty \\ Q_{i}\Delta X_{j}=M_{ij} \end{array}\right)}{Limit}Q_{i}\Delta X_{j}\frac{G(x;\alpha ,\beta )-G(x;\alpha ,\beta +\Delta X_{j})}{\Delta X_{j}}=M_{ij}\frac{\partial }{\partial X_{j}}G(x;\alpha ,\beta )).$$

Thus, the displacement field for a moment tensor component with forces acting in the *i* direction and separated in the *j* direction is given as
(22)$$u_{i}^{M_{ij}}=M_{ij}\frac{\partial }{\partial X_{j}}G(x;\alpha ,\beta )).$$

#### 4.1.1. Wavefield Due to *M*_11_ Dipole Excitation

A through-thickness *M*_11_ dipole moment excitation was considered, as presented in Figure 10 for representing the dipole moment excitation due to the present Mode-I fracture case. Therefore, for predictive modeling of the Mode-I fracture case, a closed-form solution for *M*_11_ dipole moment excitation response at the PWAS is required, which is derived in this section.

First, the velocity field due to a through-thickness force excitation was derived. Then, the velocity field due to a through-thickness dipole *M*_11_ dipole moment excitation was derived using the principle in Equation (22). Next, the strain field equation was derived from the velocity field equation. Finally, from the strain field equation, a closed-form solution for PWAS voltage response was derived using Equation (15).

##### In-Plane Line Force Excitation—Normal Mode Expansion (NME)

The wavefield, due to the dipole moment excitation, is derived from the wavefield due to the force excitation. For obtaining the through-thickness *M*_11_ moment tensor excitation field, a through-thickness force excitation, as presented in Figure 11, is considered. The limiting process of the through-thickness couple forces separated by an infinitesimal distance converts the force to a moment and the wavefield due to the force to a wavefield due to a moment, as discussed in Figure 9. The separation of the forces required for obtaining the moment *M*_11_ in Figure 10 from the force in Figure 11 is in the *x*-direction. The normal mode expansion (NME) was used for obtaining the solution for the elastodynamic field due to the through-thickness force excitation in a thin plate, as presented in Figure 11.

We let the elastodynamic state 1 represent the field due to a body force excitation, *F*_1_, acting on the control volume. Hence, we represent state 1 as the sum of normal modes.
(23)$$\begin{array}{l} v_{1}=\sum_{m}a_{m}(x)v_{m}(y)=v_{1}(x,y)\\ T_{1}=\sum_{m}a_{m}(x)T_{m}(y)=T_{1}(x,y) \end{array},$$
where $a_{m}(x)$ is the modal participation factor or normal mode expansion coefficient, which corresponds to the excitation. This coefficient needs to be determined. The summation contains all wave modes that significantly contribute to the total wave field. The notations v and T represents the velocity and stresses of the wave fields. Since $v_{1}$ and $T_{1}$ represent a single-mode component, the same set of amplitude coefficients is used for particle velocity and stress field. The normal mode expansion coefficient $a_{m}$ is a function of *x*. The wave mode amplitude consolidation for different modes changes with the nature of the excitation source. *v_m_*(*y*) is the velocity modeshape, and *T_m_*(*y*) is the stress modeshape of the normal mode.

Elastodynamic state 2 is assumed as the wave field of mode *n*, expressed as follows:(24)$$\begin{array}{l} v_{2}(x,y)=v_{n}(y)e^{-i\xi_{n}x}\\ T_{2}(x,y)=T_{n}(y)e^{-i\xi_{n}x}\\ F_{2}=0 \end{array},$$
where *v_n_*(*y*) is the velocity modeshape, *T_n_*(*y*) is the stress modeshape, and $\xi_{n}$ is the wavenumber of the Lamb wave mode.

For two elastodynamic states of the plate, the complex reciprocity relationship can be given as
(25)$$\nabla \cdot ({\tilde{v}}_{2}\cdot T_{1}+v_{1}\cdot {\tilde{T}}_{2})=-{\tilde{v}}_{2}\cdot F_{1}-v_{1}\cdot {\tilde{F}}_{2},$$
where ~ represents the complex conjugate. Expanding the “del” operator, substituting Equations (23) and (24) into Equation (25), and simplifying, we get
(26)$$\begin{array}{l} \int_{-d}^{+d}\frac{\partial }{\partial x}({\tilde{v}}_{n}(y)e^{i\xi_{n}x}\cdot \sum_{m}a_{m}(x)T_{m}(y)+\sum_{m}a_{m}(x)v_{m}(y)\cdot {\tilde{T}}_{n}(y)e^{i\xi_{n}x})\cdot \hat{x}dy+\\ {({\tilde{v}}_{n}(y)\cdot T_{1}+{\tilde{T}}_{n}(y)\cdot v_{1})\cdot \hat{y}|}_{-d}^{+d}e^{i\xi_{n}x}=e^{i\xi_{n}x}\int_{-d}^{+d}(-{\tilde{v}}_{n}(y)\cdot F_{1})dy \end{array}.$$
Upon further simplifying, we get
(27)$$4P_{nn}\frac{\partial }{\partial x}e^{i\xi_{n}x}\cdot a_{n}(x)=e^{i\xi_{n}x}\int_{-d}^{+d}({\tilde{v}}_{n}(y)\cdot F_{1})dy,$$
where *P_mn_* is the orthogonality relation in the general form represented as follows:(28)$$P_{mn}=-\frac{1}{4}\int_{-d}^{+d}{\tilde{v}}_{n}(y)T_{m}(y)+v_{m}(y)\cdot {\tilde{T}}_{n}(y)\cdot \hat{x}dy.$$

Due to the orthogonality condition of the Lamb wave modes, *P_mn_* = 0 [59] for m≠n, in the case of a propagating Lamb wave mode. Thus, only *P_nn_* exists in Equation (27).

Considering the Lamb wave propagating in a 1D plate, the lamb velocity vector is given as
(29)$${\tilde{v}}_{n}(y)=[{\tilde{v}}_{x}^{n}(y)\;{\tilde{v}}_{y}^{n}(y)].$$

In Equation (25), *F*_1_ is the body force excitation. For a through-thickness dynamic force excitation acting as shown in Figure 11, *F*_1_ can be expressed as follows:(30)$$F_{1}=[\begin{array}{cc} F_{x}\delta (x) & 0 \end{array}].$$

Simplifying Equation (27) using Equations (29) and (30), assuming the location of fatigue crack source as the origin and waveform measurement location as *x*, the closed-form expression for normal mode expansion coefficient is given as
(31)$$a_{n}^{x}(x)=\frac{F_{x}\int_{-d}^{d}{\tilde{v}}_{x}^{n}(y)dy}{4P_{nn}}e^{-i\xi_{n}(x)}.$$

Substituting the normal mode expansion coefficient into Equation (23), we get the expression for the velocity field due to a through-thickness dynamic force excitation in the *x*-direction (as shown in Figure 11) as follows:(32)$$v_{1}=v_{F_{x}}=\sum_{m}a_{m}^{x}(x)v_{m}(y)=\sum_{m}\frac{F_{x}\int_{-d}^{d}{\tilde{v}}_{x}^{m}(y)dy}{4P_{mm}}v_{m}(y)e^{-i\xi_{m}(x)}.$$
For unit force excitation, the equation becomes
(33)$$v_{F_{x}}=\sum_{m}a_{m}^{x}(x)v_{m}(y)=\sum_{m}\frac{\int_{-d}^{d}{\tilde{v}}_{x}^{m}(y)dy}{4P_{mm}}v_{m}(y)e^{-i\xi_{m}(x)}.$$

Line Moment (*M*_11_) Excitation Field

Two force excitations separated by an infinitesimal distance and the limiting process of that distance that goes to zero create a dipole moment excitation, as shown in Figure 12a. In the figure, PWAS is located at a distance of *r_c_* from the source, and *s_r_* is the radius of the PWAS. The limiting process of the separation turns the excitation to a moment excitation. The through-thickness *M*_11_ dipole moment excitation is presented in Figure 12b. In Equations (16)–(22), it was shown that the wavefield due to through-thickness dipole moment excitation can be obtained from the displacement field due to through-thickness force excitation from the limiting process, explained as follows:(34)$$u_{Q_{1}}^{M_{11}}(x)=\underset{\left(\begin{array}{l} \Delta X_{1}\to 0\\ Q_{1}\to \infty \\ Q_{1}\Delta X_{1}\to M_{11} \end{array}\right)}{lim}\left[Q_{1}\Delta X_{1}\right]\left[\frac{G(x;\alpha ,\beta )-G(x;\alpha +\Delta X_{1},\beta )}{\Delta X_{1}}\right]=M_{11}\frac{\partial G(x;\alpha ,\beta )}{\partial X_{1}}.$$

Here, the *G* is the velocity field due to the force excitation presented in Equation (33). Following the limiting process in Equation (34), the velocity field due to *M*_11_ line source is obtained from Equation (33) as follows:
(35)$$v_{M_{11}}=\frac{\partial }{\partial x}(v_{F_{x}})=M_{11}\sum_{m}-i\xi_{m}\frac{\int_{-d}^{d}{\tilde{v}}_{x}^{m}(y)dy}{4P_{mm}}v_{m}(y)e^{-i\xi_{m}(x)}.$$


The displacement field can be obtained from the velocity field as follows:
(36)$$u_{M_{11}}=v_{M_{11}}/i\omega =M_{11}\sum_{m}-i\xi_{m}\frac{\int_{-d}^{d}{\tilde{v}}_{x}^{m}(y)dy}{4P_{mm}}v_{m}(y)e^{-i\xi_{m}(x)}/i\omega ,$$
where *ω* is the angular frequency measured in radians per second. The in-plane strain is the spatial derivative of the displacement field as follows:(37)$$S_{M_{11}}=\frac{\partial }{\partial x}(u_{M_{11}})=M_{11}\sum_{m}-\xi_{m}{}^{2}\frac{\int_{-d}^{d}{\tilde{v}}_{x}^{m}(y)dy}{4P_{mm}}v_{m}(y)e^{-i\xi_{m}(x)}/i\omega .$$

The strain field is sensed by the PWAS and converted to the equivalent voltage by the PWAS. Following Equation (15), the total voltage sensed by PWAS can be obtained from the integral as follows:(38)$$V_{M_{11}}^{PWAS}=\int_{S_{c}}V_{3}=\frac{g_{31}^{2}}{\left(s_{11}^{E}\beta_{33}^{T}C_{c}\right)}\int_{S_{c}}\varepsilon_{xx}dS,$$
where *S_c_* represents the limits of integration over the PWAS. Substituting Equation (37) into Equation (38), we get
(39)$$V_{M_{11}}^{PWAS}=\frac{g_{31}^{2}}{\left(s_{11}^{E}\beta_{33}^{T}C_{c}\right)}\int_{S_{c}}M_{11}\sum_{m}-\xi_{m}{}^{2}\frac{\int_{-d}^{d}{\tilde{v}}_{x}^{m}(y)dy}{4P_{mm}}v_{m}(y)e^{-i\xi_{m}(x)}/i\omega dS.$$
Rearranging the integral term, we get
(40)$$=\frac{g_{31}^{2}}{\left(s_{11}^{E}\beta_{33}^{T}C_{c}\right)}M_{11}\sum_{m}\frac{\int_{-d}^{d}{\tilde{v}}_{x}^{m}(y)dy}{4P_{mm}}v_{m}(y)\int_{S_{c}}\xi_{m}{}^{2}e^{-i\xi_{m}(x)}dS/i\omega .$$

Suppose the PWAS is located at a distance of *r_c_* from the source. Additionally, let *s_r_* be the radius of the PWAS. Then, the limit of integration *S_c_* is from *r_c_* to 2*s_r_*. Substituting the limits of integration, we get
(41)$$=\frac{g_{31}^{2}}{\left(s_{11}^{E}\beta_{33}^{T}C_{c}\right)}M_{11}\sum_{m}\frac{\int_{-d}^{d}{\tilde{v}}_{x}^{m}(y)dy}{4P_{mm}}v_{m}(y)\int_{r_{c}}^{r_{c}+2s_{r}}\xi_{m}{}^{2}e^{-i\xi_{m}(x)}dx/i\omega .$$
Performing the integration and substituting the limits, we get
(42)$$=\frac{g_{31}^{2}}{\left(s_{11}^{E}\beta_{33}^{T}C_{c}\right)}M_{11}\sum_{m}\frac{\int_{-d}^{d}{\tilde{v}}_{x}^{m}(y)dy}{4P_{mm}}v_{m}(y)\left(\xi_{m}{}^{2}\frac{e^{-i\xi_{m}(r_{c}+2s_{r})}-e^{-i\xi_{m}(r_{c})}}{-i\xi_{m}}\right)/i\omega .$$

Rearranging the terms, we get a closed-form expression for the voltage sensed by the PWAS as follows:(43)$$V_{M_{11}}^{PWAS}=\frac{g_{31}^{2}}{\left(s_{11}^{E}\beta_{33}^{T}C_{c}\right)}M_{11}\sum_{m}\frac{\int_{-d}^{d}{\tilde{v}}_{x}^{m}(y)dy}{4P_{mm}}v_{m}(y)\xi_{m}\left(e^{-i\xi_{m}(r_{c})}-e^{-i\xi_{m}(r_{c}+2s_{r})}\right)/\omega .$$

#### 4.1.2. Mode-1 Fracture AE Simulation

The dipole moment source definition for Mode-I fatigue crack growth was used for simulation of fatigue crack growth AE. For the time domain of the dipole source excitation, a commonly used wide-band cosine bell function excitation from the literature [14] was assumed. The mathematical representation for the function is given as follows:(44)$$\begin{array}{cc} E(t)=-1 & t<0\\ E(t)=0.5(1-\cos(pi(t/\tau )))-1 & t>0\\ \tau =\mathrm{Rise}\;\mathrm{time} & \; \end{array}.$$

The flow chart for predictive simulation of AE signal propagation is presented in Figure 13. First, the Fourier transform of the cosine bell excitation function, *E*(*t*), was performed. Then, the structural transfer function *G* of the Mode-I fracture formation was calculated. A Mode-I fracture has an equivalent AE source excitation of the *M*_11_ dipole moment. Thus, the structural transfer function *G* is the closed-form solution derived in Equation (43) for a unit excitation. Then, the structural transfer function for a Mode-I fracture was multiplied with the cosine bell excitation function in the frequency domain to obtain the frequency domain of the signal at the receiver. Finally, the inverse Fourier transform of the frequency domain of the signal at the receiver was performed to obtain the AE signal prediction at the receiver PWAS.

The variables in the numerical simulation were taken from the experiment. The dimension of the PWAS used during the experiment was 7 mm. For the simulation, the PWAS dimension, 2 *s_r_*, was also taken as 7 mm. The location of the PWAS *r_c_* was taken as 25 mm. The material properties corresponding to the aluminum 2024-T3 specimen were considered (70 GPa Young’s modulus, 0.33 Poisson’s ratio, and 2780 kg/m^3^). The rise time parameter of the equation was adjusted to obtain the best similarity between the experiment and simulation. The frequency content of the excitation function varies according to the rise time, which is assumed. This, in fact, varies the frequency content of the resulting predictive simulation. The rise time was adjusted to have a good similarity between the experimental observation and simulation. The cosine bell function excitation was adjusted to a rise time of *s*, as shown in Figure 14, to match the simulation closely with the experimental observation.

The wavenumber plot for S0 and A0 Lamb wave modes for 1 mm aluminum 2024-T3 is presented in Figure 15. By using the expression in Equation (43), the normalized PWAS tuning curve for *M*_11_ moment excitation was plotted for the S0 and A0 Lamb wave mode, as shown in Figure 16. For an *M*_11_ excitation, a strong S0 mode content was observed, and no A0 mode response was excited.

Using the procedure in Figure 13 and the cosine bell function excitation in Figure 14, the simulation of fatigue crack growth AE signal was performed. The resulting predicted waveform after performing the bandpass filtering corresponding to the pre-amplifier (Section 2.1) is presented in Figure 17a. We observe the nondispersive S0 mode in the simulation. The time-frequency representation of the numerical simulation and experiment was performed using Choi-Williams transform, which is also presented in Figure 18a. A side-by-side comparison of the time-domain simulation signal and the experimental signal and the Choi-Williams transform of the simulation signal and the experimental signal is presented in Figure 17 and Figure 18, respectively. In the experimental observation, a nondispersive S0 wave packet was also observed at the beginning of the signal. However, in the experiment, apart from the initial nondispersive wave packet, the presence of a slow traveling wave packet was also observed. This wave packet is insignificant in example 2 and weakly present in example 1. This later arriving wave packet observed in the experiment is due to the scattering of the AE waves generated at the crack tip due to the crack formation at the crack and reaching the PWAS sensors following the initial wave packet. At present, in the simulation, we consider an ideal two-dimensional (2D) case in which a crack length in the third dimension is not considered. Therefore, the present simulation does not predict the scattered later weak wave packet.

### 4.2. Verification of the Analytical Method

For verification of the theoretical modeling equations, an FEM numerical analysis of the PWAS response was performed. A 400 mm long and 1 mm thick 2D model was developed using the ANSYS software package (Figure 19a). In the FEM model, the material properties corresponding to the aluminum 2024-T3 specimen were considered (70 GPa Young’s modulus, 0.33 Poisson’s ratio, and 2780 kg/m^3^). The element chosen for the finite element modeling was the plane-183 element. The AE waves due to the excitation were received using a receiver PWAS of 7 mm width and 0.5 mm thickness with the material properties of APC-850, located at 25 mm from the AE source. For the modeling of PWAS, plane-82 elements were used. For eliminating the reflections from the boundaries of the plate, nonreflective boundaries (NRBs) were applied at both ends of the model using the spring damper element COMBIN14 in ANSYS [60]. NRBs were applied, as presented in Figure 19a. The COMBIN14 elements were implemented at the top and bottom surfaces and both ends of the plate. The damping coefficients of the elements were varied gradually, starting from zero in a sinusoidal pattern on the plate. At the edges of the plate, the maximum stiffness and damping values were applied. The maximum stiffness and damping values at the edges of the plate were *K_N_* = 0, *C_N_ = LρC_p_*, *K_T_ = 0*, and *C_T_ = LρC_s_*. Here, *K* is the stiffness coefficient, and *C* is the damping coefficient. *L* is the element length (0.05 mm). *N* and *T* are the normal and tangential directions. *ρ* is the material density. *C_p_* and *C_s_* are the bulk wave speeds of the material, 6100 m/s and 3070 m/s, respectively, for the present study. The damping coefficient values at the plate top and bottom edges were varied from zero to half the maximum value in a sinusoidal pattern(1/2 *C_N_* and 1/2 *C_T_*). The length of the sinusoidal pattern was 100 mm.

Finite element meshing was performed by selecting a 0.05 mm element size for the length and thickness of the model. The FE model, after the meshing, is presented in Figure 19b. Afterward, the M_11_ dipole excitation was defined. The M_11_ dipole excitation was modeled in the FEM by using dipole forces. The modeling details of the dipole force is presented in Figure 19b. A zoom-in of the details is shown in Figure 19c. Equal and opposite nodal forces were applied to define the M_11_ dipole excitation. A cosine bell function excitation was applied as the time profile of the excitation with 0.5 µs as the rise time of the excitation (Figure 14). The finite element simulation was performed to solve the acoustic waveforms generated due to the dipole excitation. After performing the simulation, the in-plane displacement field at the thickness nodes at 25 mm from the excitation location was extracted. The thickness modeshape of the in-plane displacement at 25 mm from the excitation was plotted and presented in Figure 19d, while the thickness modeshape of the analytical simulation is presented in Figure 19f. Due to the through-thickness dipole excitation, the in-plane displacement varies symmetrically with *y*-axis. The in-plane displacement modeshape resembles the modeshape of S0 Lamb wave mode. In the analytical prediction, it was concluded that the through-thickness dipole excitation causes the generation of pure S0 mode. As we observe from Figure 16, a strong S0 mode is generated due to the through-thickness M_11_ excitation, and no A0 mode is present. Analytical and FEM modeshapes were observed to be very closely matching, as we observe from the side-by-side comparison in Figure 19d,f. This comparison verifies the validity of the analytical simulation.

Next, the FEM PWAS response simulation was compared with the analytical prediction. The resulting PWAS response after performing the filtering corresponding to the pre-amplifier bandwidth discussed in Section 2.1 is presented in Figure 20. Figure 20a presents the normalized FEM PWAS response simulation result. Figure 20b shows the normalized analytical simulation result. A very good match of the FEM simulation and analytical simulation was observed. Both simulations showed a nondispersive AE wave sensed by the PWAS in the simulation. Afterward, the Choi-Williams transform of the FEM simulation (Figure 20c) and analytical simulation (Figure 20d) was compared. The FEM simulation and analytical simulation also showed a very good match in the Choi-Williams transform comparison. The comparison of the FEM simulation and theoretical simulation of the PWAS response proved that the theoretical simulation is valid.

## 5. Summary, Conclusions, and Future Work

### 5.1. Summary and Conclusions

This paper presented analytical predictive modeling of AE signals sensed using PWAS sensors during a fatigue crack growth event. This paper started with an introduction of the instrumentation used for sensing AE signals using PWAS transducers and a study of the transformations that AE signals undergo while passing through the instrumentation. A closed-form expression for sensing of acoustic waves using the PWAS sensor was derived. Next, the paper discussed a review of an in situ fatigue experiment performed for the detection of AE signals during a fatigue crack growth in a 1 mm aluminum 2024-T3 plate. The fatigue crack growth-related AE signals detected were presented in this section. It was observed that the fatigue crack growth AE signals recorded has a strong S0 Lamb wave mode content.

After discussing the characteristics of fatigue crack growth AE signals briefly, the paper presented the analytical modeling of guided wave propagation during the fatigue crack growth event. The dipole moment excitation concept was used for modeling the fatigue crack growth AE source. Normal mode expansion of Lamb wave modes was used for obtaining a closed-form analytical expression for predicting the AE signal recorded using PWAS sensors. Later, the paper presented the numerical prediction results using the analytical expression. A 7 mm diameter PWAS was considered as the receiver, and a cosine bell excitation of 0.5 µs was chosen as the rise time of the excitation. The prediction was compared with experimentally recorded results. First, the time-domain AE signal prediction was compared with the time-domain experimental AE signals. Next, the time-frequency representation of the AE signal prediction and experimental AE signals was obtained using Choi-Williams transform, and the results were compared. The simulation result and the experimental results showed a close match with the presence of a strong S0 Lamb wave mode in the AE signal. However, in the experimental signal, after the strong S0 mode, a weak wave packet was also observed. This phenomenon is due to the scattering of the AE waves generated at the crack. While in the analytical prediction, a 1D waveguide was considered without the effect of AE signal scattering at the crack. Finally, the paper presented the verification of the analytical method using the FEM method. The dipole moment AE source modeling and simulation were performed using the FEM method. The modeshape of the waveform and the PWAS response were calculated. The analytically calculated modeshape and PWAS response were compared with the FEM modeshape and PWAS response. The results from analytical prediction and FEM prediction were very closely matching, which verified the analytical prediction.

### 5.2. Future Work

The analytical model presented in this paper is preliminary and a proof-of-concept study. Further research needs to be done to understand better the interaction of AE signals generated at the crack tips with the crack and improve the model. A three-dimensional (3D) FEM analysis in a model with crack can be performed to study the PWAS response due to fatigue crack growth. Numerical correlation analysis for the comparison of the analytical prediction and the FEM prediction result also can be performed.

## Figures and Tables

**Figure 1 sensors-20-05835-f001:**
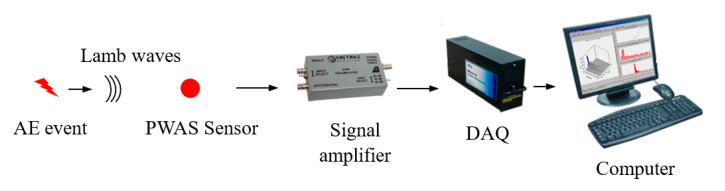
Acoustic emission (AE) signal flow diagram from initiation to digitization at the data acquisition (DAQ).

**Figure 2 sensors-20-05835-f002:**
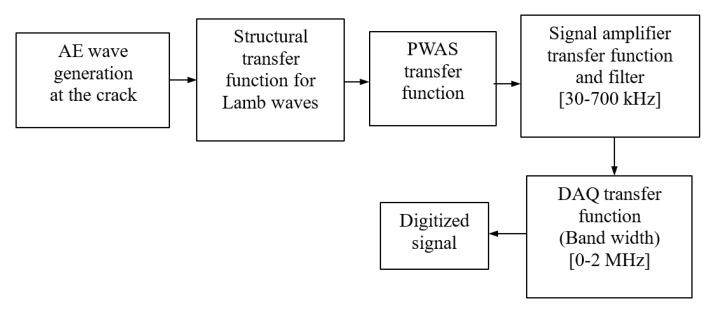
Block diagram of AE signal flow.

**Figure 3 sensors-20-05835-f003:**
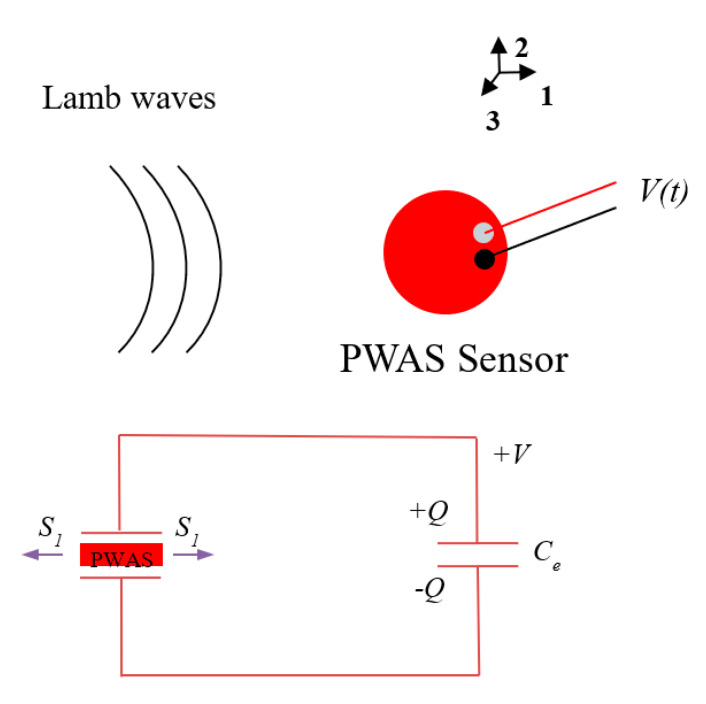
Schematic of Lamb wave sensing of piezoelectric wafer active sensor (PWAS).

**Figure 4 sensors-20-05835-f004:**
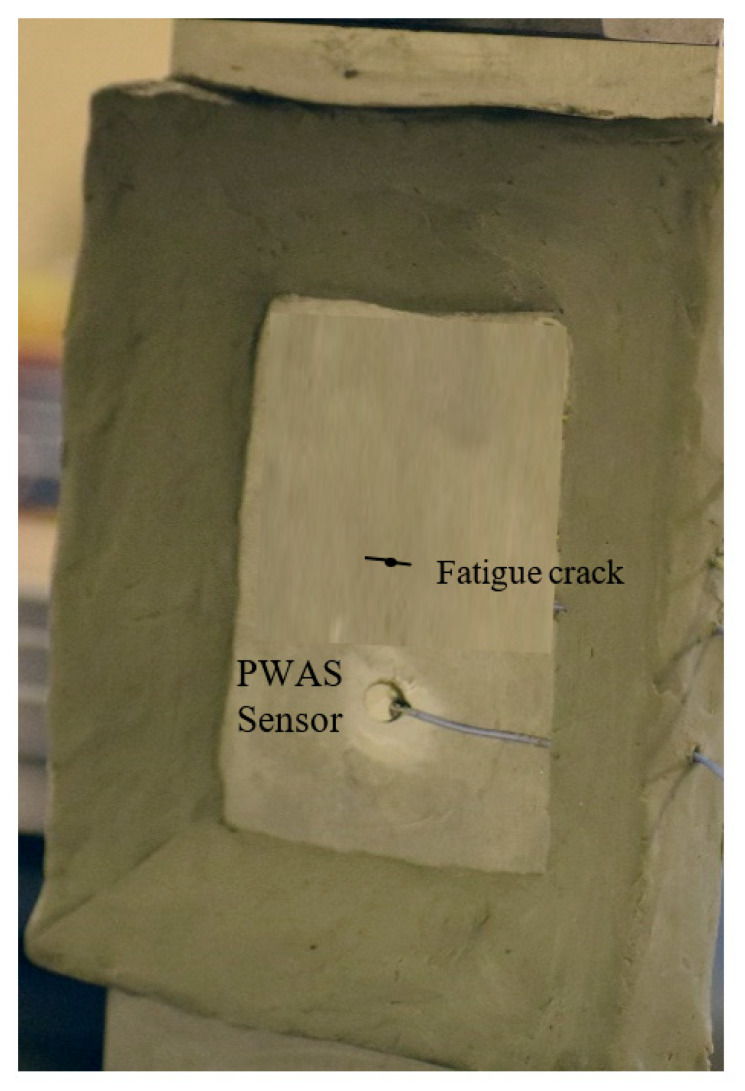
AE test specimen bonded with the PWAS sensor. Nonreflective clay boundaries (NRB) were provided on the specimen to avoid the reflection of AE signals from the specimen boundaries.

**Figure 5 sensors-20-05835-f005:**
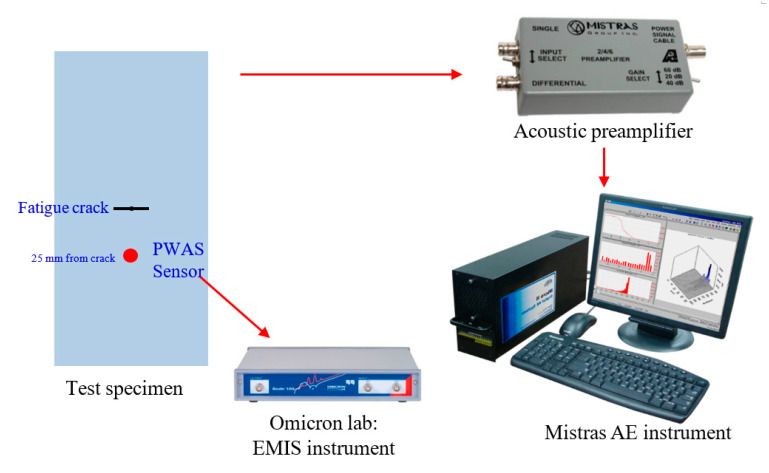
Experimental set-up for capturing AE signals during fatigue crack event.

**Figure 6 sensors-20-05835-f006:**
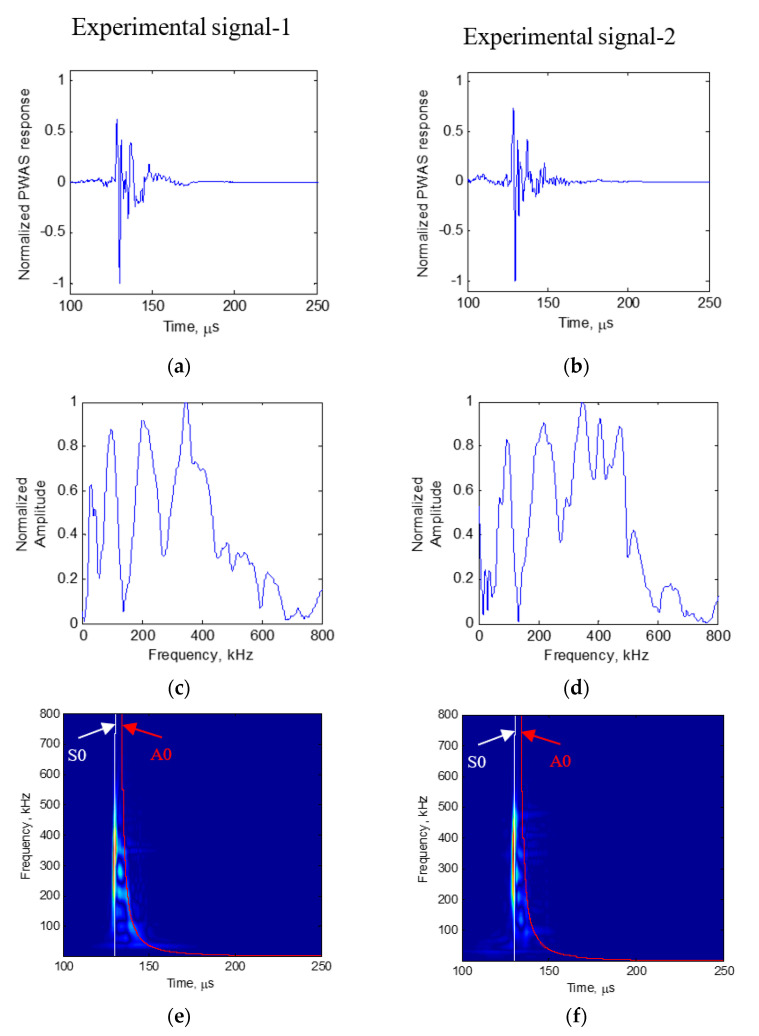
AE signals recorded during fatigue crack growth event: (**a**) Experimental AE signal-1; (**b**) experimental AE signal-2; (**c**) frequency spectrum of experimental signal-1; (**d**) frequency spectrum of experimental signal-2; (**e**) Choi-Williams transform of experimental signal-1; (**f**) Choi-Williams transform of experimental signal-2.

**Figure 7 sensors-20-05835-f007:**
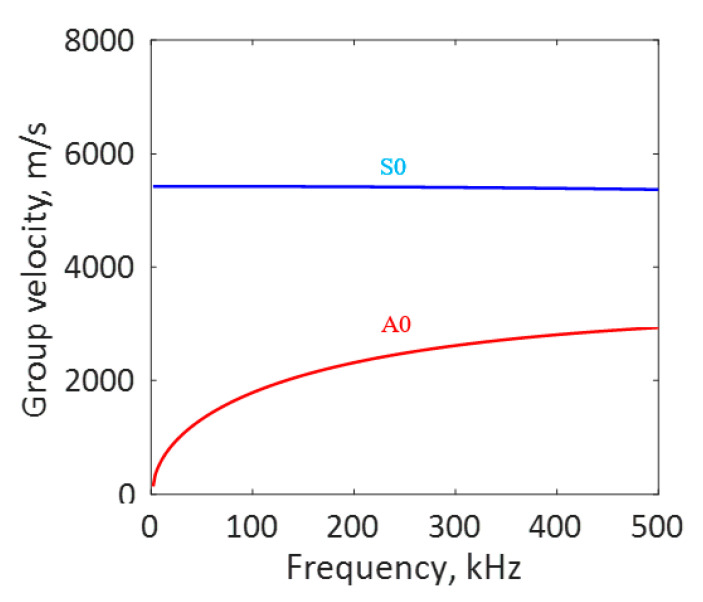
Group velocity dispersion curve of 1 mm aluminum plate.

**Figure 8 sensors-20-05835-f008:**
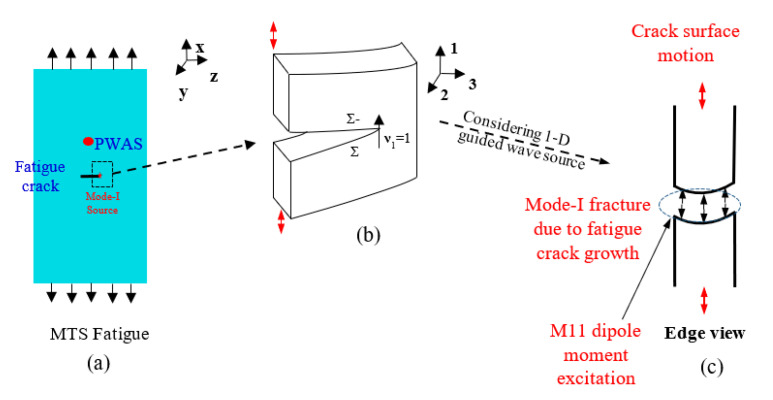
Schematic of the AE source modeling methodology: (**a**) Fatigue crack specimen schematic with loading; (**b**) schematic of zoom-in of the crack tip; (**c**) edge view of the fatigue crack tip with the schematic of moment tensor excitation corresponding to Mode-I fracture.

**Figure 9 sensors-20-05835-f009:**
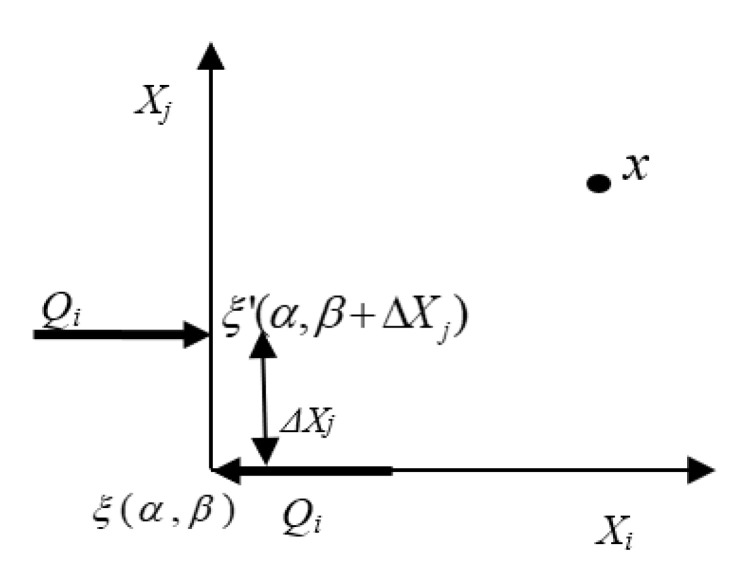
Couple force applied for generating the moment. As we limit the separation of the forces to zero, the couple becomes a concentrated moment at ξ(α,β).

**Figure 10 sensors-20-05835-f010:**
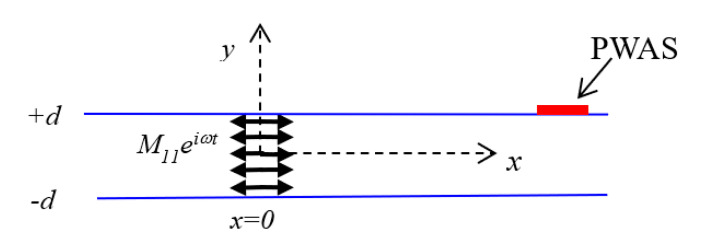
Through-thickness *M*_11_ excitation.

**Figure 11 sensors-20-05835-f011:**
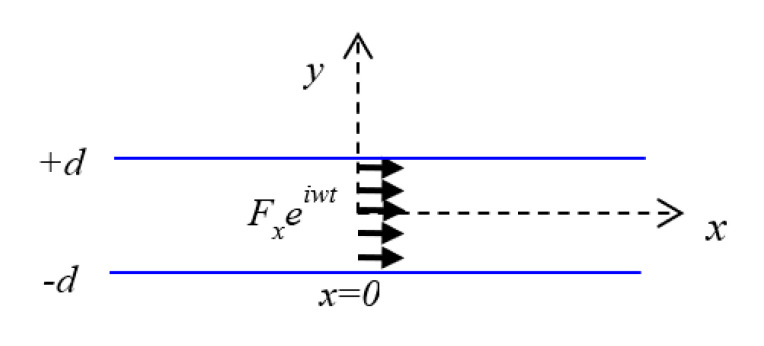
The through-thickness line force excitation in the *x*-direction.

**Figure 12 sensors-20-05835-f012:**
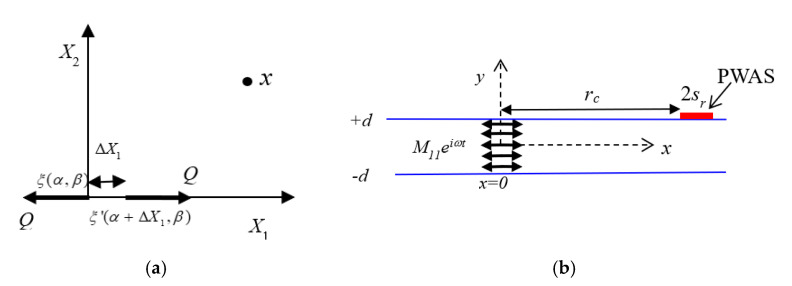
The through-thickness *M*_11_ moment excitation.

**Figure 13 sensors-20-05835-f013:**
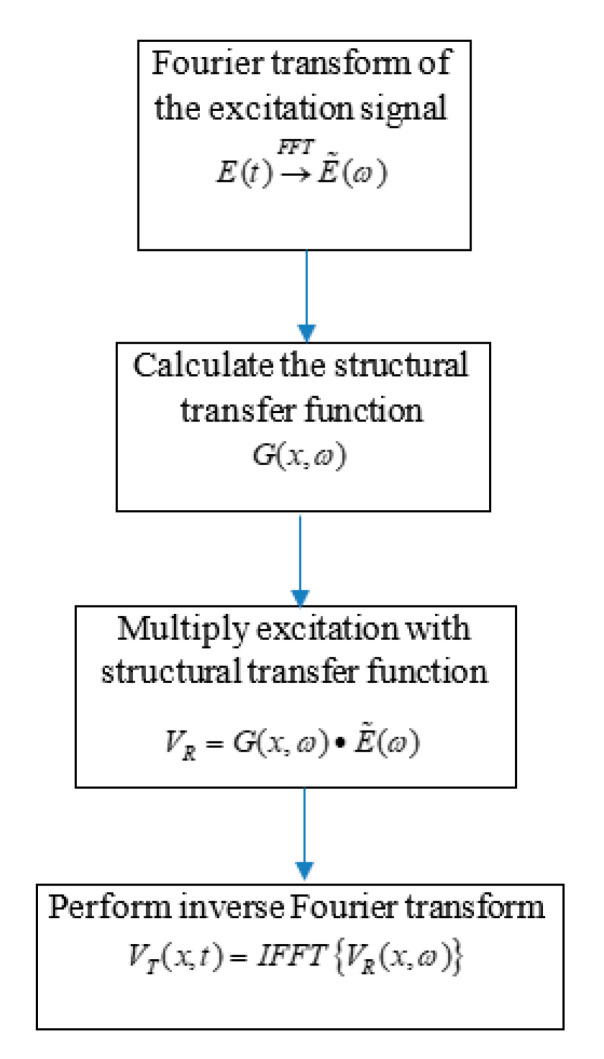
Predictive simulation of AE signal flow chart.

**Figure 14 sensors-20-05835-f014:**
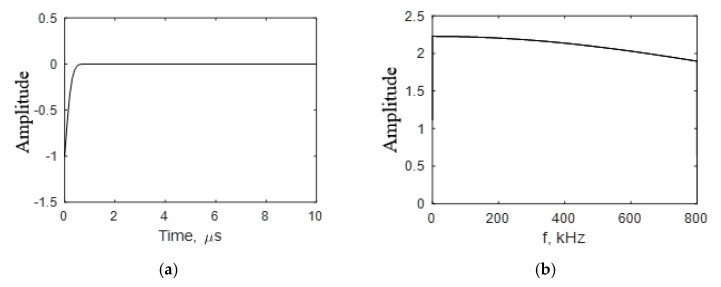
Adjusted cosine bell function excitation used for predictive modeling: (**a**) Time-domain of the cosine bell function excitation; (**b**) frequency spectrum of the cosine bell function excitation.

**Figure 15 sensors-20-05835-f015:**
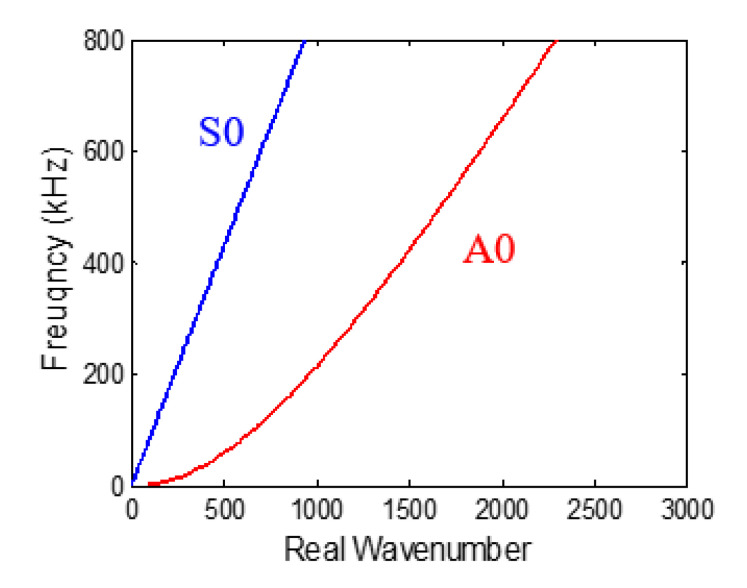
Frequency-wavenumber plot of 1 mm aluminum 2024-T3.

**Figure 16 sensors-20-05835-f016:**
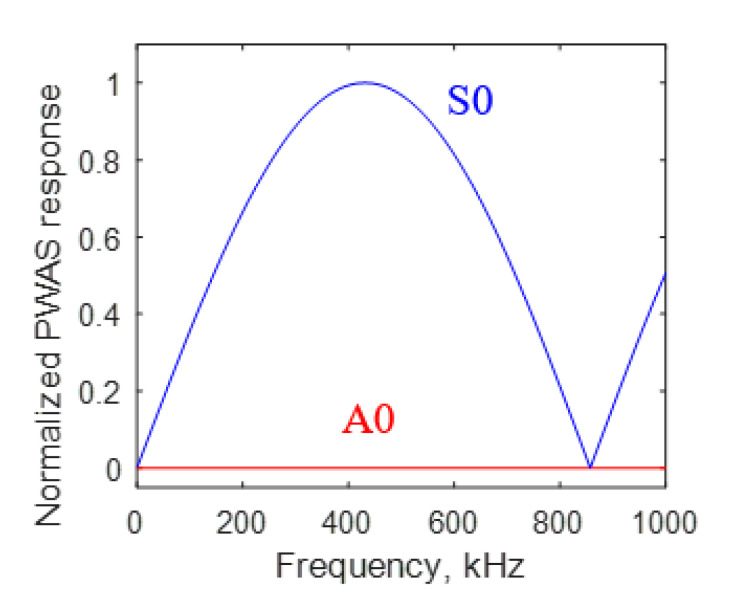
Lamb wave strain response due to *M*_11_ excitation on 1 mm thick aluminum specimen received with 7 mm PWAS receiver.

**Figure 17 sensors-20-05835-f017:**
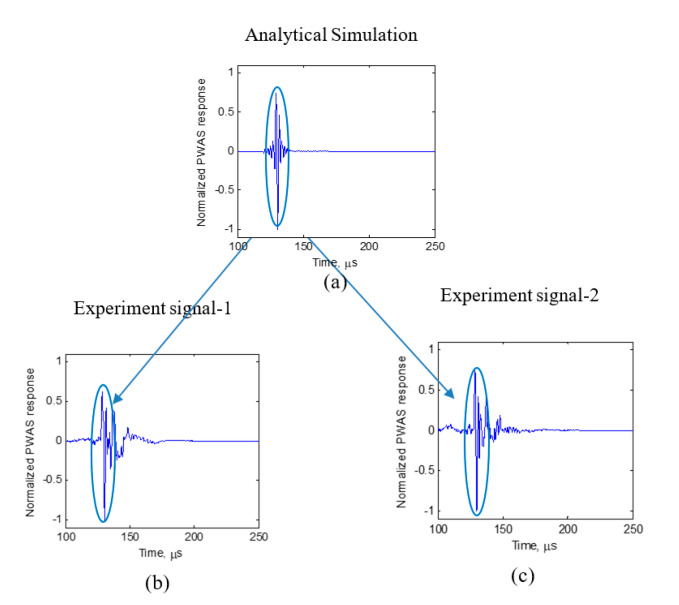
Comparison of simulation and experiment: (**a**) Simulation result; (**b**) experiment signal-1; (**c**) experiment signal-2.

**Figure 18 sensors-20-05835-f018:**
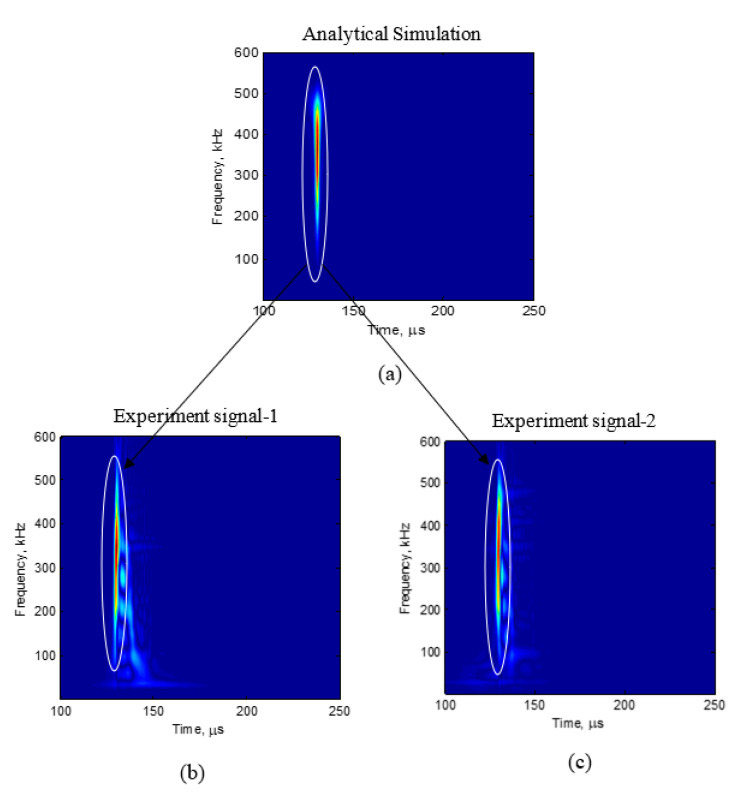
Comparison of Choi-Williams transform of simulation and experiment: (**a**) Simulation result; (**b**) experiment signal-1; (**c**) experiment signal-2.

**Figure 19 sensors-20-05835-f019:**
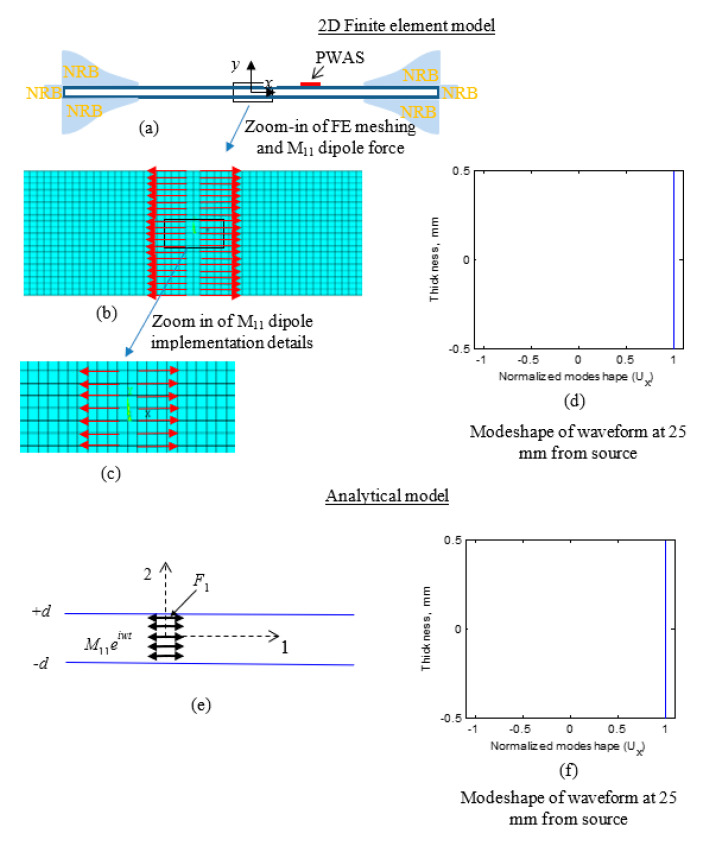
The thickness modeshape comparison of the analytical model and the finite element method (FEM) model. A very close match of FE modeshape and analytical modeshape was achieved: (**a**) 2D FEM model; (**b**) meshed FEM model; (**c**) zoom-in of meshed FEM model; (**d**) thickness modeshape of in-plane displacement from the FEM simulation; (**e**) schematic of the analytical model; (**f**) thickness modeshape of in-plane displacement from the analytical simulation.

**Figure 20 sensors-20-05835-f020:**
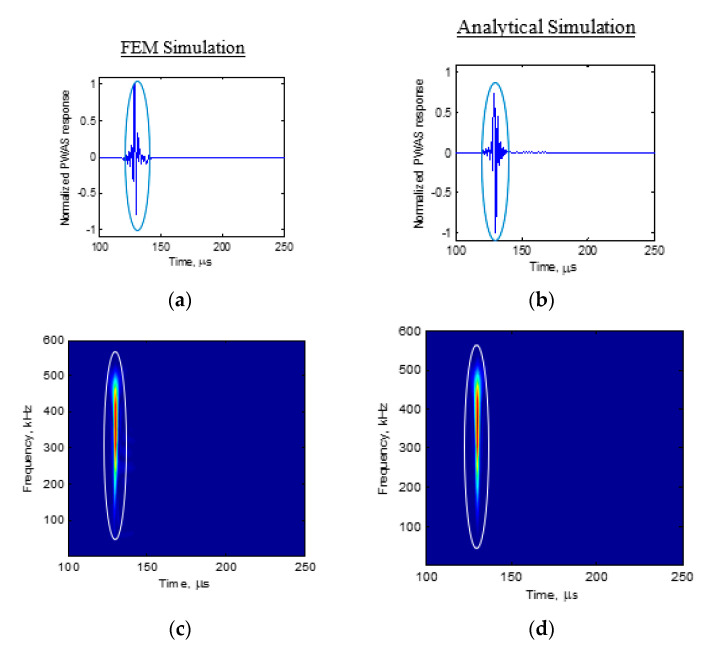
Comparison of FEM Simulation and analytical simulation: (**a**) FEM simulation waveform; (**b**) analytical simulation waveform; (**c**) FEM simulation Choi-Williams transform; (**d**) analytical simulation Choi-Williams transform.

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
