# Peer review of "Analytical and Experimental Study of Fatigue-Crack-Growth AE Signals in Thin Sheet Metals"

_sensors, 2020, doi:10.3390/s20205835_

Round 1

Reviewer 1 Report

Dear Authors,

  1. From the title, fatigue crack growth study is expected. However, nothing related to fatigue such as repeated load or crack appears in the manuscript. The title seems to be modified. 
  2. Quite many symbols were used in equations. Some without definition were used. Detailed nomenclature is needed.
  3. In some graphs, i.e. Figure 14 and 19, the axis titles were not given. All axis title should be given in consistent manners. As far as possible, use the symbols in equations as the axis titles and insert units. In Figure 19 (b) and (f), the order of thickness values is different. 
  4. Why did not you compare the amplitude between the finite element results with the analytical results as Figure 17? If possible, compare the finite element results with the analytical results with experimental results together.

Author Response

1. From the title, fatigue crack growth study is expected. However, nothing related to fatigue such as repeated load or crack appears in the manuscript. The title seems to be modified. 

Response: Thank you very much for the comment. The author agree to the reviewer’s comment that the key information of fatigue loading is missing in the text. The maximum and minimum loading applied during the experiment has been updated in the experiment section (Section 3.1, line 222-224). The crack length increment information was already present in the section 3.1 (Section 3.1, line 236). The AE signals were recorded during the fatigue crack growth as explained in the section 3 (Review of experiments). So the authors would like to retain the “fatigue-crack-growth” term in the title.

2. Quite many symbols were used in equations. Some without definition were used. Detailed nomenclature is needed.

Response: Thank you very much for the comment. The authors have updated the missing nomenclature of the symbols in page 6 (line 182-184, line 211), Page 13 (line 396) page 14 (line 414, 419), page 15 (line 436)

3. In some graphs, i.e. Figure 14 and 19, the axis titles were not given. All axis title should be given in consistent manners. As far as possible, use the symbols in equations as the axis titles and insert units. In Figure 19 (b) and (f), the order of thickness values is different. 

Response: Thank you very much for the comment. Appropriate axis titles has been given to Figure 14 and 19. The order of thickness values has been unified in Figure 19 (d) and (f)

4. Why did not you compare the amplitude between the finite element results with the analytical results as Figure 17? If possible, compare the finite element results with the analytical results with experimental results together.

Response: Thank you very much for the comment. The authors agree to the reviewer’s comment of comparing the results with FEM prediction. The PWAS response prediction also has been performed using the FEM modeling of PWAS in the plate and added to the results (Figure 20 in the manuscript). The FEM prediction and analytical results were showed good agreement as we observe from the figure and corresponding explanations in section 4.2.

Reviewer 2 Report

The manuscript describes the analytical modeling of the AE signals generated by fatigue crack growth collected by PWAS sensors.

The introduction is comprehensive and provides the reader with a good number of background works to understand the described phenomenon. Concerning FEM, the phrase "for simple as well as complicated geometries" at line 89 should be removed or replaced with something more detailed, since it does not contribute to the discussion.

The method section is also well written, with just some minor errors, like the numbering of sections (section 2.1 is duplicated) and some formulas. In particular, in equation (2), the compliance apex is D instead of E; in equation (15) the term S_c appears without being defined (it will be defined much later, on line 420).

In the review section, line 232, the use of Fahrenheit degrees is strongly discouraged by the reviewer in favor of Celsius degrees, according to the choice of using the metric SI length units throughout the entire work. Also in the discussion of the experimental results Figure 7 is cited before Figure 6: the reviewer strongly suggest moving the content of line 269 ahead of line 265. 

In the following section, concerning the in-plane force excitation, at line 383 the body force excitation is cited with reference ("here") to equation (29): equation (25) should be cited instead. On line 398, the two forces "create" the dipole moment (not "creates"). On line 410, the omega symbol is missing after the word "where". Lines 449 and 485 should be rewritten to improve quality and readability. On line 452 and 496, since the cited material properties are more than one, please replace "was" with "were". On line 475, 476, and 535, please correctly cite the surnames of the authors of the Choi-Williams transform. Finally, please extend section 4.2, since it is too brief and correct Figure 19: the spell-checker red wiggles are clearly visible. Consider numerically computing the error between the simulated and analytical waveforms instead of mere visual comparison.

In the concluding sections, please rewrite line 539 locution "considered without considering" and reduce the number of self-citation: 11 out of 60 involves papers written by the authors of this manuscript.

Some repetitions in the text reduce its quality: it is strongly suggested by the reviewer to find synonyms for "working conditions" (lines 30-32), "failure" (lines 33-34), "AE" (lines 43-51), "many researchers" (line 98), "instrumentation" (lines 142-144), "PWAS" (lines 150-152), "structures" (lines 160-161).

As a concluding remark, the work is sound and its writing is satisfying: some minor concerns must be addressed before considering this work suited for publication.

Author Response

The manuscript describes the analytical modeling of the AE signals generated by fatigue crack growth collected by PWAS sensors.

The introduction is comprehensive and provides the reader with a good number of background works to understand the described phenomenon. Concerning FEM, the phrase "for simple as well as complicated geometries" at line 89 should be removed or replaced with something more detailed, since it does not contribute to the discussion.

Response: Thank you for the comment. The corresponding sentence has been removed (page 2, line 89)

The method section is also well written, with just some minor errors, like the numbering of sections (section 2.1 is duplicated) and some formulas. In particular, in equation (2), the compliance apex is D instead of E; in equation (15) the term S_c appears without being defined (it will be defined much later, on line 420).

Response: Thank you for the comment. The numbering of section 2.1 has been revised. The compliance apex D has been updated to E in equation (2) (Page 5, line 181). The definition of Sc has been given where it appears first in equation (15) (Page 6, line 211)

In the review section, line 232, the use of Fahrenheit degrees is strongly discouraged by the reviewer in favor of Celsius degrees, according to the choice of using the metric SI length units throughout the entire work. Also in the discussion of the experimental results Figure 7 is cited before Figure 6: the reviewer strongly suggest moving the content of line 269 ahead of line 265. 

Response: Thank you for the comment. In line 232 (now line 231), the Fahrenheit degree measurement has been updated to Celsius degrees. The placement of figure 7 (now figure 6) has been changed to be after figure 6 (now figure 7). The authors also agree to the suggestion of moving the content of line 269 ahead of line 265 and has been moved in the updated manuscript (section 3.3, first paragraph).

In the following section, concerning the in-plane force excitation, at line 383 the body force excitation is cited with reference ("here") to equation (29): equation (25) should be cited instead. On line 398, the two forces "create" the dipole moment (not "creates"). On line 410, the omega symbol is missing after the word "where". Lines 449 and 485 should be rewritten to improve quality and readability. On line 452 and 496, since the cited material properties are more than one, please replace "was" with "were". On line 475, 476, and 535, please correctly cite the surnames of the authors of the Choi-Williams transform. Finally, please extend section 4.2, since it is too brief and correct Figure 19: the spell-checker red wiggles are clearly visible. Consider numerically computing the error between the simulated and analytical waveforms instead of mere visual comparison.

Response: Thank you very much for the comments. At line 383 (now 380), the equation (25) has been correctly cited. On line 398 (now 395), “creates” has been changed to “create”. On line 410 (now 408) “Omega” symbol has been updated. Lines 449 (now 453) and line 485 (now 488) has been rewritten to improve the readability. In line 452 (now 450) and 496 (now 494) “was” has been changed to “were”. In 475 (now 472), 476 (now 474), and 535 (now 536), the “Choi-Williams” transform has been correctly spelled. Section 4.2 has been expanded with more explanations. The numerical calculation of the error between the simulated and the analytical waveforms is a good idea. However, the present research, a numerical error calculation, was not focused. This suggestion has been added to future work (Section 5.2, lines 583-584).

In the concluding sections, please rewrite line 539 locution "considered without considering" and reduce the number of self-citation: 11 out of 60 involves papers written by the authors of this manuscript.

Response: Thank you very much for the comment. Line 539 (now 572) has been updated. The number of self-citations has been reduced in the manuscript. Some citations are retained for the readers advantage.

Some repetitions in the text reduce its quality: it is strongly suggested by the reviewer to find synonyms for "working conditions" (lines 30-32), "failure" (lines 33-34), "AE" (lines 43-51), "many researchers" (line 98), "instrumentation" (lines 142-144), "PWAS" (lines 150-152), "structures" (lines 160-161).

Response: “Working conditions” in lines 30-32 has been replaced by the synonym “Operation”, Failure (lines 33-44) has been replaced by “damage”, The repetition of AE in lines 43-51 has been removed by rephrasing the sentences. Many researchers (line 98) has been removed. Repetition of “instrumentation” has been avoided and replaced with “DAQ”. The repetition of PWAS in line 150-152 (now 148-149) has been replaced by “transducer”. The repetition of “structures” in lines 160-161 (now 157-158) has been avoided and has been replaced by “plate”.

As a concluding remark, the work is sound and its writing is satisfying: some minor concerns must be addressed before considering this work suited for publication.

Round 2

Reviewer 1 Report

Dear Authors,

Please see the attached review comments.
